# Total Antioxidant Capacity in Obese and Non-Obese Subjects and Its Association with Anthropo-Metabolic Markers: Systematic Review and Meta-Analysis

**DOI:** 10.3390/antiox12081512

**Published:** 2023-07-28

**Authors:** Wendoline Anaya-Morua, José Rafael Villafan-Bernal, Esther Ramírez-Moreno, Humberto García-Ortiz, Raigam Jafet Martínez-Portilla, Cecilia Contreras-Cubas, Angélica Martínez-Hernández, Federico Centeno-Cruz, Florencia Estefana Pedroza-Montoya, Lorena Orozco, Francisco Barajas-Olmos

**Affiliations:** 1Academic Area of Medicine, Interdisciplinary Research Center, Institute of Health Sciences, Autonomous University of the State of Hidalgo, Pachuca 42160, Mexico; an467887@uaeh.edu.mx; 2Investigador por México, Consejo Nacional de Humanidades, Ciencia y Tecnología (CONAHCYT), Mexico City 03940, Mexico; jvillafan@inmegen.edu.mx; 3Iberoamerican Research Network in Obstetrics, Gynecology and Translational Medicine, Mexico City 42160, Mexico; sub.invclinica@inper.gob.mx (R.J.M.-P.); florencia.pedroza@edu.uaa.mx (F.E.P.-M.); 4Immunogenomics and Metabolic Diseases Laboratory, Instituto Nacional de Medicina Genómica, SS, Mexico City 14610, Mexico; hgarcia@inmegen.gob.mx (H.G.-O.); ccontreras@inmegen.gob.mx (C.C.-C.); amartinez@inmegen.gob.mx (A.M.-H.); fcenteno@inmegen.gob.mx (F.C.-C.); 5Academic Area of Nutrition, Interdisciplinary Research Center, Institute of Health Sciences, Autonomous University of the State of Hidalgo, Pachuca 42184, Mexico; esther_ramirez@uaeh.edu.mx; 6Clinical Research Branch, Evidence-Based Medicine Department, National Institute of Perinatology, Mexico City 11000, Mexico

**Keywords:** total antioxidant capacity, obesity, metabolic parameters, metanalysis, oxidative stress biomarkers

## Abstract

The total antioxidant capacity (TAC) has been related to the development of and complications associated with chronic diseases, but its importance during obesity is not entirely clear. We conducted a systematic review and meta-analysis to clarify whether there are differences or similarities in the TAC between subjects with obesity (SO) and subjects with normal weight (NW). Following the recommendations of PRISMA and Cochrane, we performed a systematic search in the PubMed, Scopus, Web of Science, Cochrane, and PROSPERO databases, identifying 1607 studies. Among these, 22 studies were included in the final analysis, comprising 3937 subjects (1665 SO and 2272 NW) in whom serum TAC was measured, and from these 19,201 subjects, the correlation of serum TAC with anthropo-metabolic parameters was also estimated. The Newcastle–Ottawa method was used for the evaluation of the risk of bias. Using a random-effect model (REM), TAC was reduced in SO independently of age (SMD, −0.86; 95% CI −1.38 to −0.34; *p* = 0.0012), whereas malondialdehyde (SMD, 1.50; 95% CI 0.60 to 2.41), oxidative stress index (SMD, 1.0; 95% CI 0.16 to 1.84), and total oxidant status (SMD, 0.80; 0.22 to 1.37) were increased. There were seven significant pooled correlations of TAC with anthropometric and metabolic parameters: weight (r = −0.17), hip circumference (r= −0.11), visceral adipose index (r = 0.29), triglycerides (r = 0.25), aspartate aminotransferase (r = 0.41), alanine aminotransferase (r = 0.38), and uric acid (r = 0.53). Our results confirm a decrease in TAC and an increase in markers of oxidative stress in SO and underpin the importance of these serum biomarkers in obesity.

## 1. Introduction

Oxidative stress is defined as the imbalance between oxidants and antioxidants, with more antioxidants, inducing tissue injury in the body through DNA damage, lipid peroxidation, and protein denaturation, thus contributing to the pathogenesis of several conditions, including cardiovascular and metabolic diseases [1,2,3]. To counteract oxidative stress, the human body synthesizes antioxidant enzymes and employs non-enzymatic antioxidants mainly coming from dietary intake and endogenous sources [4]. Therefore, estimating the antioxidant activity in body fluids and tissue homogenates may be helpful in understanding any impairment mechanisms of this process in diseases. The total antioxidant capacity (TAC) measures the capability of a fluid or tissue to scavenge free radicals without considering the activity of oxidants and redox enzymes in such a sample [5,6]. This measurement presents advantages over other chemical methods since it integrates all measurable antioxidants and is thus a widely reported biochemical parameter. TAC has attracted attention in the study of diverse diseases such as type 2 diabetes, where low serum TAC has been related to its pathogenesis and chronic complications in both the microvascular and cardiovascular systems [7,8,9]. However, in other pathologies such as obesity, differences in serum TAC between subjects with obesity (SO) and subjects with normal weight (NW) are still controversial, as well as whether these differences are significantly related to other metabolic abnormalities [10,11]. For example, while some studies have reported lower serum TAC in SO compared to NW [10,11], others have not found such differences or have reported higher levels [12,13]. Obesity is a worldwide public health problem that contributes to excess morbidity and mortality, and there is growing evidence indicating that several of its comorbidities result from the excessive production of reactive oxygen species [14,15,16,17,18,19]. Therefore, it is necessary to look more deeply into the role of serum TAC levels in obesity and their relationship to clinical or biochemical parameters. Thus, the aim of this systematic review and meta-analysis was to determine whether differences exist in the serum TAC between SO and NW, as well as to investigate its correlation with metabolic-related traits.

## 2. Materials and Methods

### 2.1. Protocol Registration

This protocol was registered in the PROSPERO International Prospective Register of Systematic Reviews under registration number CRD42022366334.

### 2.2. Information Sources and Search Strategy

PubMed, SCOPUS, Web of Science, The Cochrane Library, and PROSPERO database were used to identify relevant studies on serum TAC in SO and NW. Other relevant publications were manually searched. The first search was conducted on 20 September 2022 by combining MeSH terminology with Boolean operators “OR” and “AND”. A screening update was extended until 30 April 2023. The included keywords were (“obeses”[All Fields] OR “obesity”[MeSH Terms] OR “obesity”[All Fields] OR “obese”[All Fields] OR “obesities”[All Fields] OR “obesity s”[All Fields]) AND ((“total”[All Fields] OR “totaled”[All Fields] OR “totaling”[All Fields] OR “totalled”[All Fields] OR “totalling”[All Fields] OR “totals”[All Fields]) AND (“antioxidant s”[All Fields] OR “antioxidants”[Pharmacological Action] OR “antioxidants”[MeSH Terms] OR “antioxidants”[All Fields] OR “antioxidant”[All Fields] OR “antioxidating”[All Fields] OR “antioxidation”[All Fields] OR “antioxidative”[All Fields] OR “antioxidatively”[All Fields] OR “antioxidatives”[All Fields] OR “antioxidizing”[All Fields]) AND (“capacities”[All Fields] OR “capacity”[All Fields])), and the search was sorted by date. A complete search strategy is provided in Appendix A. This screening was performed by two independent authors (W.A.-M. and J.R.V.-B.) who identified studies in English or Spanish without publication time restrictions. These authors filtered the relevant abstracts blinded to authorship, authors’ institutional affiliation, and study results.

### 2.3. Eligibility Criteria

This systematic review and meta-analysis adhered to the Meta-analysis of Observational Studies in Epidemiology (MOOSE) guidelines [20] and the Preferred Reporting Items for Systematic Reviews and Meta-Analyses (PRISMA) guidelines [21]. The inclusion criteria were observational studies of SO and NW undergoing the measurement of serum TAC. Studies were excluded when no information was provided on mean TAC, TAC values were reported only in graphs, no full text was available, the sample size was <20 per group, patients with obesity and overweight were combined, and the risk of bias evaluation was lower than six stars. Two reviewers independently screened each study (W.A.-M. and J.R.V.-B.), and the judgments of a third and fourth reviewer (F.B.-O. and L.O.) were employed to settle disagreements.

### 2.4. Data Extraction

Information from qualified studies was extracted by two independent researchers using a datasheet template based on one from the Cochrane Consumers and Communication Review Group. Then, two reviewers independently performed quality control on the extracted data, followed by cross-validation before statistical analysis. The information included the first author, year of publication, country where the study was conducted, inclusion and exclusion criteria, total number of patients included in the study, number of SO and NW, mean and standard deviation values for serum TAC, units and method of measurement, and intra- and inter-assay variation coefficients. When available, we collected information on age, body mass index (BMI), waist-to-hip ratio (WHR), and serum levels of metabolic parameters. We also extracted the correlation coefficient of serum TAC with anthropometric (age, weight, waist circumference, hip circumference, waist-to-hip ratio body mass index, adipose-visceral index, diastolic blood pressure, and systolic blood pressure) and biochemical parameters (levels of total cholesterol, C-reactive protein, HDL-c, LDL-c, triglycerides, aspartate aminotransferase, alanine aminotransferase, fasting glucose, HOMA-IR, insulin, uric acid, and ceruloplasmin) when available. Obesity and normal weight definitions were based on the World Health Organization or on population-specific criteria (Appendix A). These definitions and those for anthropometric and biochemical parameters were extracted from each primary article.

### 2.5. Assessment of Risk of Bias

The Newcastle–Ottawa Scale for case–control studies [22] was used to evaluate the quality of observational studies by two independent reviewers (F.E.P.M. and R.J.M.-P.). A third and fourth evaluator resolved any disagreement between the main reviewers (E.R.M. and F.C.C.). The quality of studies was judged by this scale on three dimensions: the selection of the study groups, the comparability of the groups, and the ascertainment of exposure. Only studies with six or more stars were included because they were considered high-quality [22].

### 2.6. Data Analysis

Extracted data on serum TAC levels were pooled in a meta-analysis expressing the effect size as the standardized mean difference (SMD) between SO and NW because each study reported it in different units. A pooled correlation coefficient was estimated to analyze correlations between TAC and anthropometric and biochemical parameters. We estimated the random-effect model (REM) weighting based on the inverse of variance for both SMD and pooled correlations because studies were performed in individuals of different populations, and we cannot assume that the studies share a common effect size [23]. The SMD in serum TAC between SO and NW and the pooled correlations are presented using Forest plots. Publication bias and systematic heterogeneity were visually assessed through funnel plots and the COPAS method.

Between-study heterogeneity was evaluated using τ^2^, Cochran’s Q, and I^2^ statistics [24]. A Baujat analysis was performed to evaluate the contribution of individual-study heterogeneity to the overall effect size. To assess the “small-study effect”, a cumulative analysis was performed and presented as a Forest plot.

To explain heterogeneity and determine variables affecting serum TAC, we performed analysis by subgroups and multiple meta-regression with the Hedges method as planned in the PROSPERO protocol, using patient age, altitude, and the continent as covariates. We added other covariates such as BMI, because its values were wide-ranging within SO; the method for TAC quantification, because the introduction of different kits over the years based on different principles might have an impact on heterogeneity; and the sample size, because at a lower sample size, there is a higher dispersion of data according to the central limit theorem. The resulting R^2^ was reported to represent the amount of heterogeneity explained by the model and regression coefficients to stand the influence of each covariate.

The complete statistical analysis was conducted using R studio v4.2.1 (The R Foundation for Statistical Computing, https://www.R-project.org/, accessed on 8 August 2022) and the packages meta v6.5-0, metafor v4.2-0, and metasens v1.5-2.

## 3. Results

### 3.1. Study Selection

A total of 1607 studies were found: 1605 during database searching and 2 manually. Figure 1 shows the PRISMA flow diagram. The full texts of 52 studies were assessed for eligibility: 22 were included in the qualitative and quantitative synthesis (Table 1), the reasons for non-inclusion of the 30 studies are summarized in Table 2, and the complete list is shown in Appendix A. The most frequent reasons for the exclusion of studies were the inclusion of overweight subjects within SO or NW groups, the co-existence of comorbidities such as type 2 diabetes or polycystic ovarian syndrome in the OS group, a sample size lower than 20 by group, and no comparison of SO versus NW.

**Table 1 antioxidants-12-01512-t001:** Summarized characteristics of the included studies (n = 3937 subjects). Complete characteristic list is shown in Appendix A.

Author (Year of Publication)	Country of the Sample	Sample Size (n)			Proportion of SO:NW	Mean Age (Years) at Inclusion		Mean BMI		Proportion of Male:Female
		SO	NW	Total		SO	NW	SO	NW	
Amirkhizi (2010) [25]	Iran	25	79	104	0.24:0.76	39	27	33.6	23.4	0:1
Asghari (2021) [26]	Iran	140	90	230	0.61:0.39	41	40.2	32.3	23.6	0.58:0.42
Aslan (2017) [27]	Turkey	27	26	53	0.51:0.49	30	28	36.31	36.31	0.58:0.42
Aysegül (2014) * [28]	Turkey	38	51	89	0.43:0.57	9.42	9.29	27.63	17.42	0.52:0.48
Chen (2014) & [11]	China	31	30	61	0.51:0.49	44.9	44.03	28.9	22.7	1:0
Chrysohoou (2007) [10]	Greece	540	1226	1766	0.31:0.69	50	41	-	-	0.40:0.60
Dambal (2011) [29]	India	50	50	100	0.5:0.5	-	-	-	-	0.40:0.60
Dursun (2016) [30]	Turkey	20	20	40	0.5:0.5	27.9	26.11	34.08	22.15	0:1
Eren (2014) [12]	Turkey	95	56	151	0.63:0.37	13.34	13.95	31.14	18.85	0.43:0.57
Faienza (2012) † [31]	Italy	55	30	85	0.65:0.35	11.4	10.4	2.22	0.37	0.51:0.49
García-Sánchez (2020) [32]	Mexico	33	23	56	0.59:0.41	56.45	68.7	-	-	0.25:0.75
Hadžović-Džuvo (2015) [33]	Herzegovina and Bosnia	23	36	59	0.39:0.61	-	-	-	-	0:1
Karaouzene (2011) [34]	Algeria	85	120	205	0.41:0.59	48	46	33.2	23.5	1:0
Lejawa (2021) [13]	Poland	49	49	98	0.5:0.5	30.5	30.9	32.6	23.36	1: 0
Mahasneh (2016) [35]	USA	35	46	81	0.43:0.57	-	-	-	-	0:1
Matusik (2015) * † [36]	Poland	78	82	160	0.49:0.51	13.96	13.72	2.96	0.38	0.52:0.48
Park (2016) & [37]	Korea	33	45	78	0.42:0.58	65.3	66.6	26.3	22.5	0.55:0.45
Pirgon (2013) [38]	Turkey	46	29	75	0.61:0.39	12.5	12.7	30.63	18.36	0.48:0.52
Rowicka (2017) + [39]	Polonia	62	21	83	0.75:0.25	7.5	6.4	23.5	19.5	0.39:0.61
Skalicky (2008) [40]	Czech Republic	40	48	88	0.45:0.55	50	52.12	35.3	21.86	0.52:0.48
Sonoli (2015) [41]	India	70	35	105	0.67:0.33	23.1	22.9	31.38	23.07	0.5:0.5
Vehapoglu (2016) * [42]	Turkey	90	80	170	0.53:0.47	7.4	7.2	25.78	16.75	0.47:0.53

& Cut-off of BMI population specific to obesity; + the cut-off for obesity was BMI with z-score ≥ 3 standard deviations; * The criterion for obesity was BMI ≥ the 95th. † Values are expressed as BMI Z-score. SO: Subjects with obesity; NW: normal-weight subjects; BMI: body mass index.

**Table 2 antioxidants-12-01512-t002:** Summary of reasons for non-inclusion in this metanalysis.

Reason for Non-Inclusion	Number of Articles
The inclusion of overweight subjects within the SO group	5
The inclusion of overweight subjects within the NW group	4
All included patients had type 2 diabetes	4
The sample size was lower than 20 by group	4
No comparison of SO versus NW, instead there was a comparison of metabolic syndrome versus non-metabolic syndrome	4
Studies performed in patients with polycystic ovary syndrome	2
Article written in Turkish	1
Duplicated publication, data were published previously	1
Low score on the Newcastle–Ottawa scale	1
TAC was measured in seminal fluid	1
The sample size was lower than 20 by group and Newcastle–Ottawa score was 5.	1
The study groups were prediabetes versus normal fasting glycemia; thus, patients were not classified as SO and NW	1
No comparison of TAC in SO versus NW, instead authors compare men versus women	1

SO: Subjects with obesity; NW: normal-weight subjects; BMI: body mass index; TAC: total antioxidant capacity.

**Figure 1 antioxidants-12-01512-f001:**
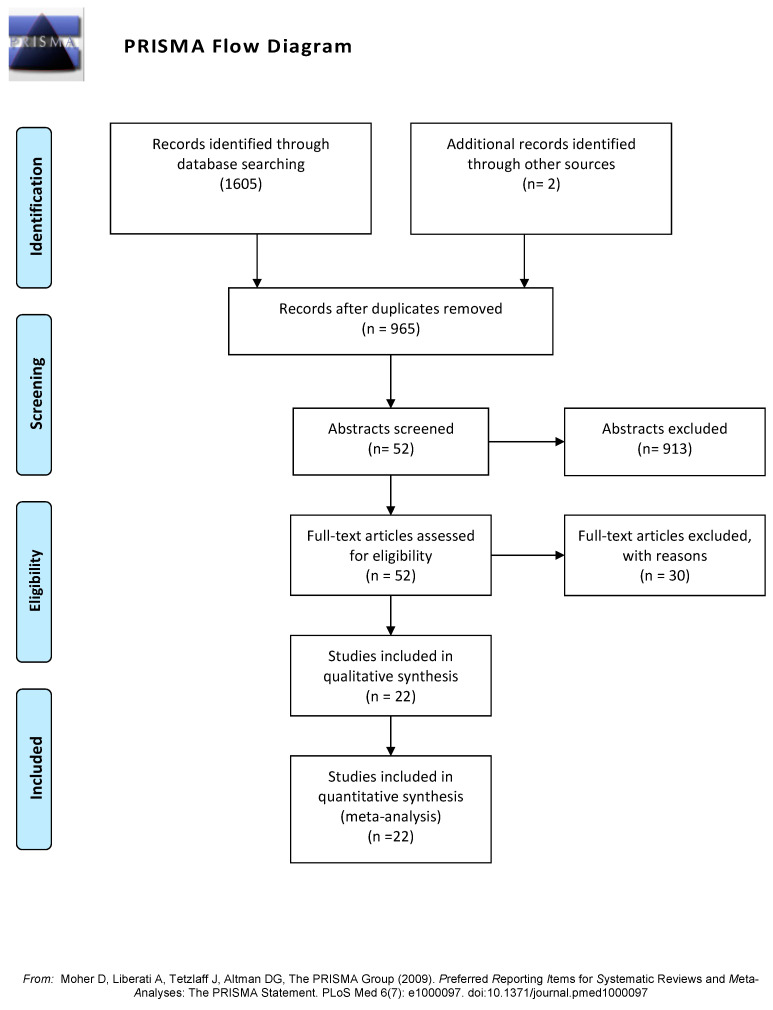
PRISMA flow diagram showing the four phases of the study (search, screening, eligibility and inclusion steps) [43].

### 3.2. Risk of Bias of the Included Studies

In order to improve on future studies, we assessed the risk of bias using the Newcastle–Ottawa scale; one study was awarded six stars, two with seven, five with eight, and fourteen with nine. The main reasons for assigning fewer stars were a lack of representativeness of the cases, no explicit selection of the controls, and a lack of control for additional outcomes (Table 3).

### 3.3. Synthesis of the Studies

This meta-analysis comprised 3937 subjects (1665 SO and 2272 NW) from 22 studies. The mean age of SO was 30.6 ± 18.5 years, and that of NW was 29.8 ± 19.4 years. The mean BMI was 30.4 ± 4.1 Kg/m^2^ for SO and 21.0 ± 2.6 Kg/m^2^ for NW. The pooled male-to-female proportion was 0.56:0.44. All studies quantified TAC in serum using a colorimetric method, and the intra- and inter-assay coefficients of variation oscillated between 3 and 7%. The selected studies’ sample sizes ranked 21 to 1226 in NW groups and 21 to 540 in the SO groups. Seven studies were performed in Europe, twelve in Asia, one in Africa, and two in America.

### 3.4. Differences in Total Antioxidant Capacity among SO and NW

The quantitative synthesis of the 22 included studies demonstrated lower serum TAC in SO than in NW (SMD, −0.86; 95% CI −1.38 to −0.34; *p* = 0.0012; random-effects model). The Q value (757.5; *p* < 0.0001) demonstrated that the effect size varies across studies, and the I^2^ indicates that 96.0% of such variation is attributed to the true effect rather than random error (Figure 2).

### 3.5. Correlations of TAC with Anthropometric and Metabolic Parameters

In the 22 included studies, the authors reported 21 correlations between TAC and other clinical and biochemical parameters, comprising 19,201 subjects. Among these, the most frequently correlated parameter was BMI (n = 15), followed by WHR (n = 11) and age (n = 6). After correlation analysis using the REM, we documented seven significant pooled correlations of TAC with weight (r = −0.17), hip circumference (r = −0.11), visceral adipose index (r = 0.29), triglycerides (r = 0.25), aspartate aminotransferase (r = 0.41), alanine aminotransferase (r = 0.38), and uric acid (r = 0.53). In addition, HDL-c and LDL-c had borderline correlations, and all correlations are shown in Table 4.

### 3.6. Subgroup Analysis by Sex, Age, Methods Used for Determining Serum TAC, and Study Sample Size

To explain the heterogeneity between studies and determine the characteristics influencing effect size, we performed a subgroup analysis. Pooled results showed significant differences in TAC between female SO and NW (SMD, −1.56; 95% CI −3.08 to −0.004) but not males (SMD, −1.00; 95% CI −2.51 to 0.52; Figure 3).

Significant differences in serum TAC between SO and NW remained in adult and pediatric patients (SMD −0.98, 95% CI −1.71 to −0.25; and −0.60, 95% CI −1.09 to −0.11, respectively; Figure 4). Subgroup analysis revealed that in studies with a sample size >100, the SMD of serum TAC was significantly different between SO and NW (SMD, −1.35; 95% CI −2.24 to −0.46) but not in studies with n < 100 (Figure 5). Furthermore, the studies based on the reduction of Cu^+2^ to Cu^+1^ for measuring TAC exhibited opposite results to the rest of the methods and the highest variability of the results (Appendix A). Therefore, sex, sample size, and the method of TAC measurement influence the differences in serum TAC but do not explain the whole heterogeneity between studies.

### 3.7. Other Oxidative-Stress-Related Parameters and Their Correlation with Anthropo-Metabolic Parameters

Other parameters related to oxidative stress that were included were arylesterase, catalase, superoxide dismutase, ceruloplasmin, zinc and copper redox in red blood cells, glutathione peroxidase, malondialdehyde (MDA), oxidative stress index (OSi), paraoxonase, total oxidant status (TOS), vitamin C, and vitamin E (Appendix A). The metanalysis of these parameters revealed that MDA (SMD, 1.50; 95% CI 0.60 to 2.41), OSi (SMD, 1.0; 95% CI 0.16 to 1.84), and TOS (SMD, 0.80; 95% CI 0.22 to 1.37) were significantly higher in SO than NW (Appendix A).

During the systematic review of articles, we found two out of three studies reported a positive correlation of MDA with weight, BMI, waist circumference, and waist-to-hip ratio. From eight studies measuring TOS, four communicated positive correlations with BMI, visceral adipose index, waist-to-hip ratio, fat mass, obesity duration, triglycerides, total cholesterol, ApoB, HDL-c, hsCRP, glucose, HbA1c, and uric acid but negative correlation with fat-free mass and predicted muscle mass. Finally, from five studies, including OSi, three revealed a positive correlation with BMI, HOMA-IR, and insulin and a negative correlation with visceral adipose index and triglyceride serum levels (Appendix A).

### 3.8. Heterogeneity, Variability Analysis, and Publication Bias

The multivariate meta-regression was employed to simultaneously analyze the effect of some cofactors on the heterogeneity (sex, the continent of origin of patients, BMI, altitude, and the method for TAC quantification).

The meta-regression results show that the only significant cofactor explaining variability was the method of the reduction of Cu^+2^ to Cu^+1^ for TAC measurement (estimates: −68.09; 95% CI −0.83.36 to −52.82; *p* < 0.0001) without residual heterogeneity (I^2^ = 0) (Table 5).

Funnel plot analysis showed a trend toward a small-study effect in which studies with higher standard errors (smaller studies) represented a significant variation in the pooled estimates (Appendix A). Also, COPAS analysis showed a very low probability that only studies with a large number data could have been published, which means a low probability of publication bias (Appendix A).

## 4. Discussion

### 4.1. Main Findings

Cumulative evidence suggests that the alteration of TAC is relevant to health. The evidence of good-quality studies demonstrates that TAC is reduced in subjects with obesity with a large effect size (SMD, −0.86; 95% CI −1.38 to −0.34; *p* = 0.0012) according to Cohen’s classification [44]. This finding clarifies that serum antioxidant status is altered during obesity.

The subgroup analysis of the difference in TAC between SO and NW indicated that the reduction in serum TAC is not influenced by age because the differences are conserved in pediatric and adult subjects. However, sex, sample size, and the method of TAC quantification influenced the results because only studies performed on women, including >100 subjects and whose serum TAC was measured using a different method to that based on the reduction of Cu^+2^ to Cu^+1^, led to significant differences in serum TAC between SO and NW. Therefore, the absence of significant differences in TAC in some studies and the high heterogeneity may be explained by these parameters, although we cannot discard biological variabilities or other factors not measured in the primary studies, such as the antioxidant capacity of the diet. Scientific evidence suggests that nutritional behaviors, supplement consumption, and dietary intakes can modify TAC [45,46]. For example, a recent systematic review revealed that folic acid supplementation significantly improves the antioxidative defense system by increasing TAC [47].

Our analysis also revealed a significant correlation between TAC and serum levels of uric acid, triglycerides, aspartate aminotransferase, and alanine aminotransferase. Among these, the highest correlation was found for uric acid (r = 0.53), followed by liver enzymes. These findings are plausible because previous research has documented that uric acid accounts for 60% of the antioxidant capacity of the plasma [48] and that its serum levels directly correlate with liver fat and the amount of visceral adipose tissue [49]. Uric acid has antioxidant properties in hydrophilic environments such as plasma, where it prevents lipid peroxidation and scavenges peroxynitrite and other prooxidant molecules [50]. On the other hand, the positive correlation of TAC with triglycerides is consistent with in vivo studies where TAC is positively correlated with LPL activity (r = 0.979), an enzyme whose activity increases in the presence of hypertriglyceridemia [51]. As far as we know, the physiological mechanism involving the positive correlation of TAC with liver enzymes levels has not yet been elucidated, so more studies are needed to obtain more insights into this oxidative process.

Except for TAC, no other biomarkers of antioxidant status were significantly different between SO and NW. However, three indicators of oxidant status (MDA, TOS, and OSi) were increased in SO and were correlated with adiposity, lipid, and glycemic-related traits. Our results reveal an imbalance in redox status in SO that can be measured in serum or plasma. This finding is essential since the redox imbalance is potentially involved in adipocyte dysfunction in SO and contributes to altered lipid and glucose metabolism [52].

### 4.2. Clinical Implications

Our results confirm that the serum’s capacity to neutralize oxidative stress is altered in obesity, correlating with serum levels of uric acid, liver fat, and visceral adipose tissue. These findings indicate the necessity of implementing strategies to enhance the ability to neutralize oxidative stress in obesity, given the significant evidence linking it to comorbidities such as insulin resistance, diabetes, cardiovascular disease, and complications [53,54].

### 4.3. Strengths and Limitations

There are five main strengths of this study: it included all high-quality studies available in the literature fulfilling the inclusion criteria; it clarifies that serum TAC is different between SO and NW; it identifies significant correlations of TAC, MDA, TOS, and OSi with anthropometric and metabolic parameters; it identifies the factors influencing the differences in TAC between SO; and it identifies a factor that explained 100% of the heterogeneity of the results. However, there are some limitations to this study. The main one is the methodological heterogeneity associated with the design of the primary studies and the employment of different measurement techniques for determining TAC. Furthermore, under-represented populations such as Africans and Americans are desirable since studies were performed predominately in Asians and Europeans. Future studies should consider the weakness of primary studies and the limitations identified here to improve our knowledge about the participation of TAC in obesity and to fill the gaps in this field.

## 5. Conclusions

This study documents that serum TAC is significantly lower in SO compared to those with NW; however, indicators of oxidative stress (MDA, TOS, and OS) are increased. These indicators of redox status are related to metabolic and anthropometric parameters, including HbA1c, HOMA-IR, triglycerides, uric acid, visceral adipose index, and liver enzymes. These results also demonstrate that sex and sample size can influence TAC results. Based on these findings, we advise health policies focusing on increasing TAC through dietary intake to mitigate the impact of obesity-related comorbidities such as metabolic syndrome, insulin resistance, type 2 diabetes, and cardiovascular disease.

## Figures and Tables

**Figure 2 antioxidants-12-01512-f002:**
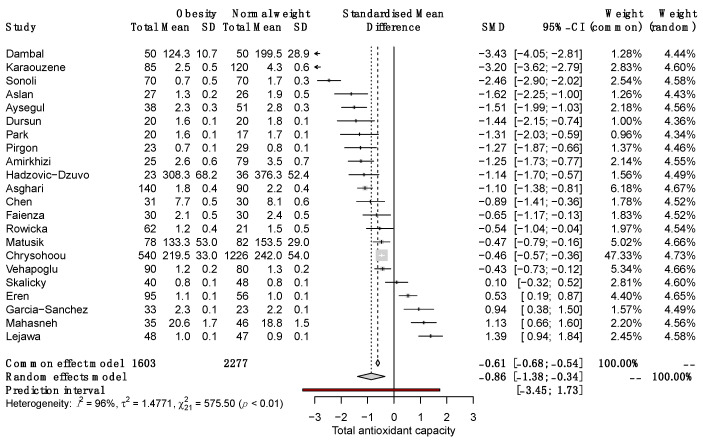
Forest plot on the difference in TAC between SO and NW for each study and the pooled estimates. SD, standard deviation; SMD, standardized mean difference; CI, confidential interval [10,11,12,13,25,26,27,28,29,30,31,32,33,34,35,36,37,38,39,40,41,42].

**Figure 3 antioxidants-12-01512-f003:**
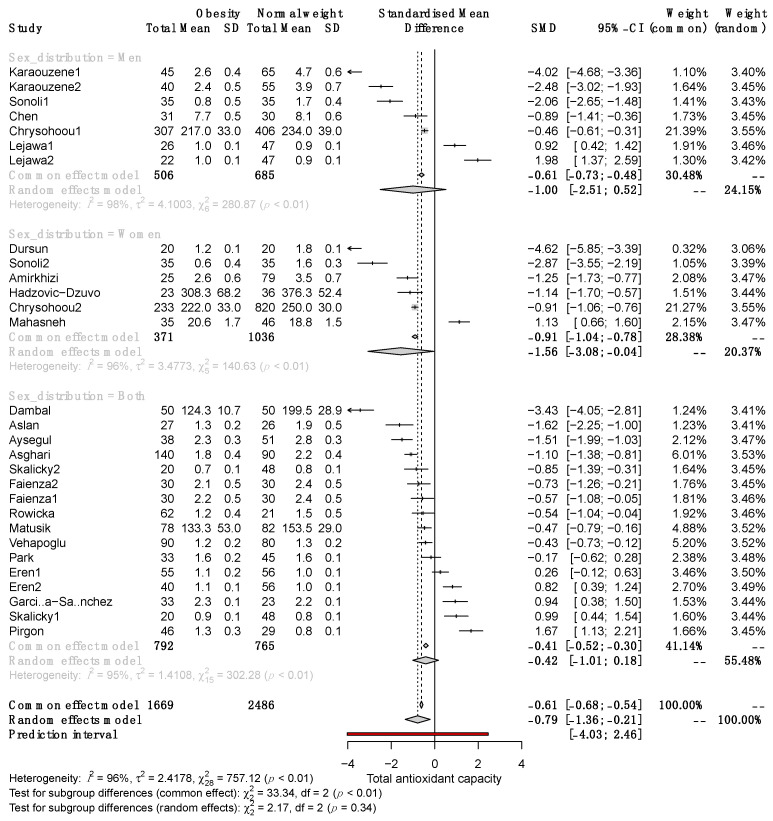
Forest plot exhibiting a subgroup analysis of the difference in TAC between SO and NW for men, women, and both sexes. SD, standard deviation; SMD, standardized mean difference; CI, confidence interval [10,11,12,13,25,26,27,28,29,30,31,32,33,34,35,36,37,38,39,40,41,42].

**Figure 4 antioxidants-12-01512-f004:**
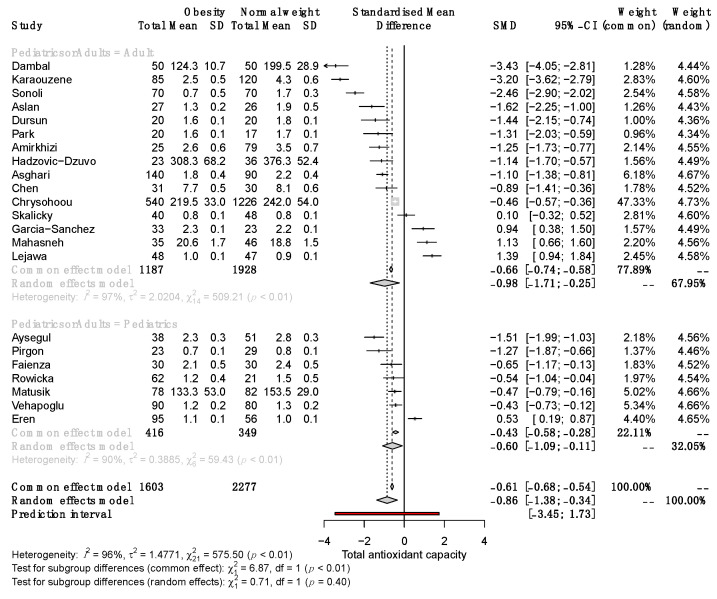
Forest plot exhibiting a subgroup analysis of the difference in TAC between SO and NW for studies performed in adult and pediatric patients. SD, standard deviation; SMD, standardized mean difference; CI, confidential interval [10,11,12,13,25,26,27,28,29,30,31,32,33,34,35,36,37,38,39,40,41,42].

**Figure 5 antioxidants-12-01512-f005:**
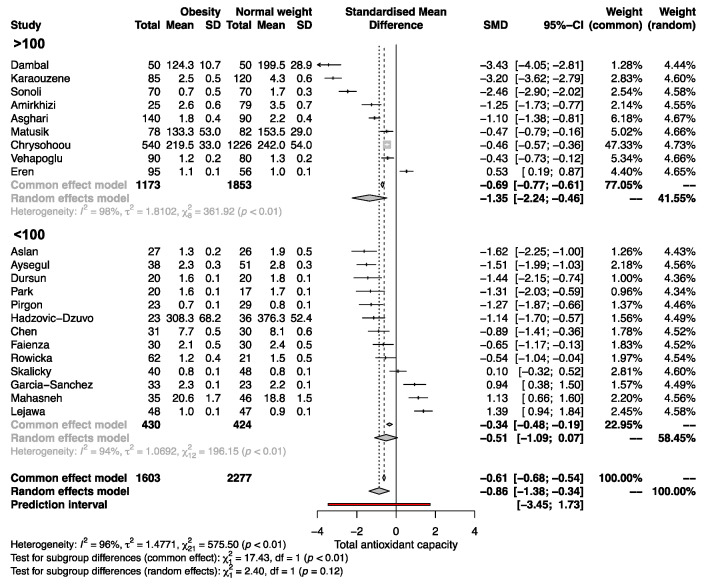
Forest plot exhibiting a subgroup analysis of the difference in TAC between SO and NW for studies including more and less than 100 subjects. SD, standard deviation; SMD, standardized mean difference; CI, confidential interval [10,11,12,13,25,26,27,28,29,30,31,32,33,34,35,36,37,38,39,40,41,42].

**Table 3 antioxidants-12-01512-t003:** Newcastle–Ottawa scale for case–control studies included in this article.

Study		Selection				Comparability			Exposure		Stars
Author	Year	Is the Case Definition Adequate?	Representativeness of the Cases	Selection of Controls	Definition of Controls	Comparability of Cases and Controls on the Basis of the Design or Analysis (Age/Other)		Ascertainment of Exposure	Same Method of Ascertainment for Cases and Controls	Non-Response Rate	
Asghari [26]	2021	*	*	*	*	*	*	*	*	*	9
Chen [11]	2014	*	*	*	*	*	*	*	*	*	9
Dambal [29]	2011	*	-	*	*	-	-	*	*	*	6
Dursun [30]	2016	*	-	*	-	*	*	*	*	*	7
Eren [12]	2014	*	*	*	*	*	*	*	*	*	9
Faienza [31]	2012	*	*	*	*	*	*	*	*	*	9
Karaouzene [34]	2010	*	*	*	*	*	*	*	*	*	9
Lejawa [13]	2021	*	*	*	*	*	*	*	*	*	9
Matusik [36]	2015	*	*	*	*	*	*	*	*	*	9
Aslan [28]	2017	*	*	*	*	*	*	*	*	*	9
Park [37]	2016	*	*	*	*	*	*	*	*	*	9
Pirgon [38]	2013	*	*	*	*	*	*	*	*	*	9
Rowwicka [39]	2017	*	*	*	*	*	-	*	*	*	8
Skalicky [40]	2008	*	*	*	*	*	*	*	*	*	9
Sonoli [41]	2015	*	*	*	*	*	-	*	*	*	9
Amirkhizi [25]	2010	*	*	*	*	*	-	*	*	*	8
Chrysohoou [10]	2007	*	*	*	*	*	-	*	*	*	8
García Sánchez [32]	2020	*	*	*	*	*	*	*	*	*	9
Hadžović-Džuvo [33]	2015	*	*	*	*	*	-	*	*	*	8
Mahasneh [35]	2016	*	*	*	*	*	*	*	*	*	9
Vehapoglu [42]	2016	*	*	*	*	*	-	*	*	*	8
Ayşegül [28]	2014	*	*	*	*	-	-	*	*	*	7

*: star; - : not-awarded with a star.

**Table 4 antioxidants-12-01512-t004:** Correlations of TAC with anthropometric and metabolic parameters (n = 19,201).

Parameter	n	Pooled Correlations (Random Effects Model)	I^2^
Age	718	0.01 (−0.11; 0.13)	61%
**Weight**	**1870**	**−0.17 (−0.27; −0.06)**	**78%**
Waist circumference	1929	−0.10 (−0.32; 0.13)	84%
**Hip circumference**	**1766**	**−0.11 (−0.21; −0.01)**	**77%**
Waist-to-hip ratio	2805	−0.03 (−0.18; 0.12)	87%
BMI	3166	0.03 (−0.11;0.17)	89%
**Adipose-visceral index**	**520**	**0.29 (0.03; 0.51)**	**86%**
Total cholesterol	678	0.13 (−0.03; 0.27)	71%
C-reactive protein	351	0.19 (−0.15; 0.49)	91%
HDL-c	678	−0.16 (−0.31; 0.01)	74%
LDL-c	678	0.11 (−0.00; 0.21)	48%
**Triglycerides**	**678**	**0.25 (0.02; 0.45)**	**85%**
**Aspartate aminotransferase**	**173**	**0.41 (0.28; 0.53)**	**0%**
**Alanine aminotransferase**	**173**	**0.38 (0.25; 0.50)**	**0%**
Fasting plasma glucose	678	0.05 (−0.03; 0.12)	0%
HOMA-IR	245	0.32 (−0.63; 0.89)	98%
Insulin	245	0.28 (−0.61; 0.86)	98%
**Uric acid**	**520**	**0.53 (0.26; 0.72)**	**90%**
Diastolic blood pressure	595	0.05 (−0.14; 0.24)	78%
Systolic blood pressure	595	0.13 (−0.15; 0.38)	89%
Ceruloplasmin	140	−0.12 (−0.28; 0.05)	0%
Total	19,201		

Bold parameters denote statistical significance. BMI: Body mass index; HDL-c: High-density lipoprotein cholesterol; LDL-c: Low-density lipoprotein cholesterol; HOMA-IR: Homeostatic Model Assessment for insulin resistance.

**Table 5 antioxidants-12-01512-t005:** Multivariate meta-regression analysis of heterogeneity modulators.

Covariate	Estimate	95% CI		*p*-Value	R^2^ (%) *
Sex female	0.0986	1.5050	1.7023	0.9040	100.0
Asian continent	1.4377	−0.4697	3.3451	0.1396	
European continent	1.8341	−0.0275	3.6957	0.0535	
BMI	−0.0315	−0.1857	0.1227	0.6890	
Age	−0.0003	0.0150	0.0144	0.9698	
Altitude	−0.0001	−0.0029	0.0001	0.7924	
Method of TAC quantification: reduction of Cu^+2^ to Cu^+1^	−68.0922	−83.3645	−52.8199	<0.0001 **	
Method of TAC quantification: Enzymatic reaction of peroxides and peroxidases (TBM)	−0.5555	1.4932	−2.6042	0.5951	
Intercept	−0.9830	−5.3560	3.3901	0.6595	

* Other estimates: Tau^2^ = 4081.0152, tau = 63.8828, H^2^ = 50. CI, confidential interval, ** denote statistical significance.

## Data Availability

Publicly available data were analyzed in this study.

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
