# Peer review of "Total Antioxidant Capacity in Obese and Non-Obese Subjects and Its Association with Anthropo-Metabolic Markers: Systematic Review and Meta-Analysis"

_antioxidants, 2023, doi:10.3390/antiox12081512_

Round 1

Reviewer 1 Report

The present systematic review has reported that serum TCA levels are significantly lower in obese subjects compared to normal-weight subjects. This conclusion was reached through a meta-analysis of 22 previous clinical studies. Overall, the study is attractive, well-designed, and well-written. However, there are a few minor errors that should be revised. As follows;

1. English should be checked by native speakers at least two. 

2. Since TAC is important, why not list and analyze the methods used for determing serum TAC in the 22 included papers.

3. Generally, TAC has been always co-determined with other oxidative stress related parameters, such as ROS, SOD, GSH, Catalase, GSH-px etc. It is important to include these data for comparison with TAC. This will help us determine if TAC is more closely correlated with metabolic markers than other oxidative stress parameters, providing a measure of its sensitivity.

Reviewer 2 Report

The authors conducted a systemic review and meta-analysis of observational studies to assess the differences in serum total antioxidant capacity levels between obese and normal-weight individuals. The authors showed that obese individuals had lower serum total antioxidant capacity levels on average than normal-weight individuals. Overall, the manuscript is well-written. In addition, the topic of this review and the evidence synthesized is essential. There are some comments.

Comments:

1.      Introduction (Lines 44-47): "Oxidative stress defines the imbalance between oxidants and antioxidants towards the first ones, inducing tissue damage in the body through DNA damage, lipid -." This statement describes the essential roles played by oxidative stress in multiple diseases. However, the reference cited describes oxidative stress's importance in colorectal cancer. More supportive references should be cited.

2.      Introduction: This study focused on obesity. However, based on the description in Introduction, it is unclear why the authors focused on obesity. A clearer explanation of the rationale, for instance, oxidative stress's role in obesity, is suggested.

3.      Materials and Methods (Eligibility criteria): The authors described the inclusion and exclusion criteria. However, it is unclear how many reviewers screened each publication and whether they worked independently.

4.      Materials and Methods (Data extraction): The authors described the information collected for each qualified study. However, some details of the data extraction process were unclear. For instance, it is unclear how many investigators reviewed each study, whether the investigators reviewed independently, and whether there were any processes for obtaining and confirming data from authors of reviewed studies.

5.      Materials and Methods (Data extraction): Please clearly describe the following variables' definitions that were used in the data extraction process: obesity (SO), normal weight (NW), serum total antioxidant capacity (TAC), intra- and inter-assay variation coefficients, weight, waist circumference, hip circumference, body mass index, waist-to-hip ratio, adipose-visceral index, diastolic blood pressure, and systolic blood pressure.

6.      Materials and Methods (Data analysis): According to the registered protocol (CRD42022366334) of this study, multiple meta-regressions were performed using the following covariates: patient age, altitude where the study was conducted, and sex. However, according to the manuscript's description, the covariates also included BMI and year of publication. It is unclear why authors finally decided to consider BMI and publication year in investigating the possible causes of heterogeneity. A description of the rationale is recommended.

7.      Materials and Methods (Data analysis): Please describe the subgroup analysis. Moreover, regarding the subgroup analysis, according to the registered protocol, it was stated that "if possible, a subgroup analysis will be conducted to evaluate the influence of gender, age groups, comorbidities or altitude on the main outcome." However, according to the manuscript, subgroup analysis by different sex, age groups, and sample sizes was conducted. Why was subgroup analysis by comorbidities or altitude not performed? Why was subgroup analysis by sample size performed? A description of the rationale is recommended.  

8.      Abstract: Please describe the methods applied in assessing the risk of biases

9.      Titles: This is a review of observational studies. Please avoid using terms implying an effect, such as "impact on." An example of a more appropriate alternative would be "association with."
